# High-Quality Assembly and Comparative Analysis of *Actinidia latifolia* and *A. valvata* Mitogenomes

**DOI:** 10.3390/genes14040863

**Published:** 2023-04-03

**Authors:** Wangmei Ren, Liying Wang, Guangcheng Feng, Cheng Tao, Yongsheng Liu, Jun Yang

**Affiliations:** 1College of Horticulture, Anhui Agriculture University, Hefei 350002, China; 2Key Laboratory of Bio-Resource and Eco-Environment of Ministry of Education, College of Life Sciences, Sichuan University, Chengdu 610017, China

**Keywords:** *Actinidia latifolia*, *Actinidia valvata*, mitogenome, phylogenetic analysis

## Abstract

Kiwifruit (*Actinidia*) has been recently domesticated as a horticultural crop with remarkably economic and nutritional value. In this study, by combining sequence datasets from Oxford Nanopore long-reads and Illumina short-reads, we de novo assembled two mitogenomes of *Actinidia latifolia* and *A. valvata*, respectively. The results indicated that the *A. latifolia* mitogenome has a single, circular, 825,163 bp molecule while the *A. valvata* mitogenome possesses two distinct circular molecules, 781,709 and 301,558 bp, respectively. We characterized the genome structure, repeated sequences, DNA transfers, and dN/dS selections. The phylogenetic analyses showed that *A. valvata* and *A. arguta*, or *A. latifolia* and *A. eriantha*, were clustered together, respectively. This study provides valuable sequence resources for evolutionary study and molecular breeding in kiwifruit.

## 1. Introduction

Kiwifruit is a perennial deciduous plant with a climbing or straggling growth habit and has been recently domesticated as an economically and nutritionally important crop with an exceptionally high content of vitamin C in its fruit. The genus *Actinidia* contains about 54 species with high genetic and morphological diversity [1], among which there are large natural variations in vitamin C content and tolerance against abiotic adversities. *A. latifolia* fruit was recorded to have the highest vitamin C content among the 54 species [2]. Accordingly, it has been employed to decipher the molecular mechanism of vitamin C metabolism and transcription regulation [3,4]. Recently, its high-quality sequences of complete chloroplast genome [5] and telomere-to-telomere genome [6] were assembled.

In addition, kiwifruit cultivation has been severely affected by abiotic adversities (such as drought, salt, waterlogging, and temperature stresses), leading to yield losses and reduced fruit quality. Fortunately, *A. valvata* was reported to be tolerant to drought, salt, and waterlogging and has been used as potential resistant resources for kiwifruit breeding or rootstock selection [7,8,9]. Several transcription factors including MYB, ERF, AP2-EREBP, bHLH, WRKY, and NAC identified from *A. valvata* were implicated to be associated with its tolerance in response to waterlogging stress by integrating RNA-seq and comparative transcriptome analysis [10,11], while an CBL-interacting protein kinase (AvCIPK11) was shown to enhance the salt tolerance through promoting ROS scavenging ability and maintaining ion homeostasis [12]. An elite waterlog-tolerant rootstock cultivar KR5 was developed from *A. valvata* germplasm [9,13] and shown to enhance the tolerance of the scion to waterlogging stress [13,14]. The complete chloroplast genome of *A. valvata* has been sequenced and assembled as well [15].

In addition to the nucleus, plant cell contains two distinct cytoplasmic genetic compartments, plastids and mitochondria. These organelles are postulated to be derived from prokaryotic endosymbiont ancestors. The organelle genomes usually have a uniparental manner of transmission, and most mitogenomes of land plants are maternally inherited. 

Organelle genomes are required for sustaining organisms’ growth and development. Similar to the nuclear genome, plant organelle genome has evolved various strategies to repair DNA damage in maintaining the integrity of its genetic material to endure the damage resulting from genotoxic stresses [16]. Organelle genomes have been extensively studied to elucidate the phylogeny and evolutionary classification. Numerous complete cpgenomes and mitogenomes have been released in the Organelle Genome database (https://www.ncbi.nlm.nih.gov/genome/browse, accessed on 2 December 2022). However, the number of sequenced mitogenomes is much fewer than that of cpgenomes possibly due to the complex structures of mitogenome, which may result from the violent redox reactions with a consequence of DNA rearrangements [17]. Mitochondria are essential organelles acting as metabolic hubs and signaling platforms, involved in a variety of pivotal cellular processes such as ATP production, fatty acid oxidation, phospholipid synthesis, reactive oxygen species (ROS) generation, and maintenance. Mitochondria also confer cellular adaptation against stressed conditions, such as oxidative stress, ER stress, nutrient deprivation, and DNA damage [18]. Interestingly, plant mitogenome has evolved a cytoplasmic male sterility system that has been extensively utilized in breeding hybrid to explore the heterosis [19].

Plant mitochondrial genomes (mitogenome) manifest tremendous evolutionary diversity in size, content, and structure, intracellular gene transfer (IGT), and interspecific horizontal gene transfer [20]. However, synonymous substitution rates of mitochondrial protein-coding genes are relatively more conservative as compared to that of chloroplast and nuclear genomes [21].

In this study, we have de novo assembled two distinct mitogenomes of *A. latifolia* and *A. valvata*, respectively. For the first time we investigate the mitogenomes of the two species, with comprehensive analyses on the gene content, repeated sequences, phylogenetic relationship by comparison with other mitogenomes of *Actinidia*, and RNA-editing sites. We also inspected the phylogeny of *Actinidia* species by incorporating the cpgenomes datasets. This study provides valuable information on taxonomic classification, molecular evolution and breeding of *Actinidia* species.

## 2. Materials and Methods

### 2.1. Plant Materials and Sequencing

The tested materials of *A. latifolia* and *A. valvata* were originally collected from You County, Hunan Province (113° E, 27° N), and Tongshan county, Hubei Province (114° E, 29° N), respectively, which are preserved in Lizhuang Ecological Garden (Hefei; Anhui; China). High-quality total genomic DNA from fresh leaves was extracted using a DNA extraction kit (Tiangen Biotech, Co., Ltd., Beijing, China). The DNA library was constructed and sequenced using previously reported procedures [22].

### 2.2. Mitogenome Assembly and Annotation

The total DNA was sequenced using the Nanopore platform (PromethION, Oxford Nanopore Technologies, Oxford, UK) and Illumina Hiseq2500 platform (Illumina, San Diego, CA, USA). The two mitogenomes of the Oxford Nanopore long-reads were assembled using SMARTdenovo with default parameters [23]. To guarantee the accuracy of the mitogenomes sequence, we employed minimap2/miniasm to polish the Illumina short-reads. The Illumina reads used to polish ONT reads were accomplished by Illumina HiSeq 2500, and the raw reads have been submitted to NCBI (*A. latifolia*:SRR23984908, *A. valvata*:SRR23984899). By PE150 (paired-end 150 bp) sequencing, we obtained 109,419,604 and 108,959,662 reads in *A. latifolia* and *A. valvata*, respectively. We applied BWA (v0.1.19) [24], SAMtools (v0.1.19) [25], Racon (v1.4.20) [26], and pilon (v1.23) [27] to map the Nanopore long-read data to the assembled mitogenomes. The mitogenomes were annotated by using the online Geseq tool (https://chlorobox.mpimp-golm.mpg.de/geseq.html, accessed on 2 December 2022) with *A. arguta* as the reference mitogenome from GenBank:MH559343 [28]. The circular maps of the mitogenomes were plotted using Ogdraw [29]. The information of NGS reads and the platforms is provided in Appendix A. In addition, we have submitted the assembled sequences of *A. latifolia* and *A. valvata* to GenBank with accession Nos. OQ198584 and OQ259535-OQ259536, respectively. 

### 2.3. Analysis of Repeat Sequences and Chloroplast-to-Mitochondrion-DNA Transfer

The SSRs (simple sequence repeats) were analyzed by MISA (https://webblast.ipk-gatersleben.de/misa/, accessed on 1 January 2023) with parameters as ‘1-10 2-5 3-4 4-3 5-3 6-3’ [30]. Tandemly repeated sequences were detected using Repeats Finder v4.09 software (http://tandem.bu.edu/trf/trf.submit.options.html, accessed on 2 January 2023) with default parameters [31]. Dispersed repeats were predicted using REPuter (http://tandem.bu.edu/trf/trf.submit.options.html, accessed on 2 January 2023) with parameters as ‘Hamming Distance 3, Maximum Computed Repeats 5000, Minimal Repeats Size 30’ and filtered using an e-value cut-off of 1 × 10^−5^ [31].

The *A. latifolia* cpgenome (NC_051880.1) and *A. valvata* cpgenome (NC_050357.1) were obtained from the NCBI Organelle Genome Resources Database (https://www.ncbi.nlm.nih.gov/genome/browse#!/organelles/, accessed on 4 January 2023). In order to identify the transferred DNA fragments from chloroplast to mitochondrion genome, we used BLASTN with the following screening criteria: matching rate ≥ 70%, E-value ≤ 1 × 10^−6^, and length ≥ 40 [32]. The results were presented using the advanced circos module in Tbtools [33].

### 2.4. Phylogenetic Analyses

For phylogenetic analyses, the shared genes of the mitogenomes sequences were extracted and concatenated by Phylosuite (v1.2.1) [34] and were subjected to alignment using MAFFT (v7.450) [35]. The phylogenetic trees of the cpgenome and the mitogenome were constructed using the maximum likelihood (ML) method via RAxML (v8.1.5) with 1000 bootstrap replicates [36]. The RAxML pipeline was “raxmlHPC-PTHREADSSSE3 -f a -N 1000 -s vr.phy -n vrtree -m GTRGAMMA -x 551,314,260 -p 551,314,260 -o o1, o2 -T 10”. Then, the phylogenetic trees were visualized using the web iTOL (https://itol.embl.de, accessed on 5 January 2023) [37].

### 2.5. Substitution Rate Calculation Analysis

We detected the dN/dS ratios for 14 protein-coding sequences derived from the mitogenomes using PAML (v4.9) [38]. We selected the yn00 module to predict the nonsynonymous substitution rate (dN) and synonymous substitution rate (dS) using parameters as ‘verbose = 0, icode = 0, weighting = 0, common f3 × 4 = 0, ndata = 1′. A schematic diagram of pairwise dN/dS values was plotted by the R package ggplot2.

## 3. Results

### 3.1. Characteristics of the Mitogenomes of Actinidia valvata and A. latifolia

The *A. latifolia* mitogenome was assembled into a single, circular, 825,163 base pair (bp) molecule while the *A. valvata* mitogenome was assembled into two distinct circular molecules, 781,709 and 301,558 bp, respectively (Figure 1).

The mitochondrial genome of *A. latifolia* or *A. valvata* comprises five ATP synthase genes (*atp1*, *atp4*, *atp6*, *atp8*, and *atp9*), nine NADH dehydrogenase genes (*nad1*, *nad2*, *nad3*, *nad4*, *nad4L*, *nad5*, *nad6*, *nad7*, and *nad9*), four cytochrome C biogenesis genes (*ccmB*, *ccmC*, *ccmFc*, and *ccmFn*), three cytochrome C oxidase genes (*cox1*, *cox2*, and *cox3*), four large subunits of ribosome proteins’ genes (*rpl2*, *rpl5*, *rpl10*, and *rpl16*), ten small subunits of ribosome proteins’ genes (*rps1*, *rps2*, *rps3*, *rps4*, *rps7*, *rps10*, *rps12*, *rps13*, *rps14*, and *rps19*), a transport membrane protein’ genes (*mttB*), a maturase (matR), a ubiquinol cytochrome C reductase (cob), and two respiratory genes (*shd3* and *sdh4*). A total of 22 tRNAs and three rRNAs were annotated in the *A. latifolia* mitogenome (Appendix A).

### 3.2. Comparative Analyses of Mitochondrial Genomes in Actinidia

The complete mitogenome sequences derived from seven *Actinidia* species were retrieved by searching the mitochondrial genome resources available in the NCBI of (Appendix A). Large variation exists in the genome size, ranging from 482,544 to 1,083,267 bp; however, their GC contents are similar, ranging from 42.0% to 46.2%. We noticed that multiple copies of some core protein-coding genes (PCGs) are present after re-annotation of these mitogenomes, i.e., four out of seven mitogenomes with re-annotated multiple-copy PCGs.

We also compared collinearity between the available mitochondrial genomes of Actinidia to assess genome rearrangement between different lineages. Using *A. valvata* and *A. latifolia* mitogenomes as references, the dot-plot analyses displayed various lengths of syntenic stretches across all the tested species, with the lowest (less than 20 kb) between *A. valvata* and *A. latifolia*, as well as the highest (about 30 kb) between *A. chinensis* and *A. valvata* (Figure 2). This result indicated that the mitogenomes underwent extensive rearrangements and the mitochondrial genomic synteny decayed with time from *Actinidia* species divergence.

### 3.3. Repeat Analysis

An important feature of plant mitogenomes is the tremendous abundance of their repeated sequences in different sizes and number. These repeats are classified into large repeats (>500 bp), intermediate-size repeats (50–500 bp), and small repeats (<50 bp) [18]. Microsatellites are also defined as simple sequence repeats (SSRs), including mono-, di-, tri-, tetra-, or penta-repeat DNA [39]. Totally, 55 and 946 SSRs were predicted in the *A. latifolia* and *A. valvata*, respectively (Figure 3A,B; Appendix A). In the *A. latifolia* and *A. valvata*, the majority of SSRs possess a single-nucleotide repeated unit, particularly A/T, and the number of A/T repeated units takes up 72.72% of all identified SSR repeats. Nevertheless, the SSRs are evenly distributed in the tested mitogenomes. There are 40, 11, and 4 SSRs in *A. latifolia* with mono-, di-, and tri-repeat units, respectively. By contrast, there are 800, 95, 4, 46, and 1 SSRs in *A. valvata* with mono-, di-, tri-, tetra-, and penta-repeat units, respectively. In *A. valvata*, most SSRs in its mitogenome contain a single-nucleotide repeat unit, accounting for 84.56% of total repeats. In addition, 23 or 120 tandem repeats were identified in the *A. latifolia* or *A. valvata* mitogenomes, respectively (Appendix A). These repeats could be further investigated for their potential use in DNA fingerprinting.

Dispersed repeats have been implicated in generating genetic diversity with considerable contributions to genome evolution [40]. Four types of dispersed repeats were described, i.e., forward repeats, repeats with reverse directions, complementary repeats, or palindromic repeats [41]. In the mitogenome of *A. latifolia* or *A. valvata*, all four types of dispersed repeats were identified. In both mitogenomes, the most abundant repeats are forward repeats, accounting for 57.6% in *A. latifolia* and 52.5% in *A. valvata* of the total repeats, in which the longest fragments are 3.427 bp from *A. latifolia* and 8.503 bp from *A. valvata* (Figure 3; Appendix A).

### 3.4. Sequence Similarity between the Mitogenome and the Cpgenome

Sequence similarity analysis indicated that a total length of 18.849 or 22.7482 bp sequences identified in *A. latifolia* or *A. valvata* mitogenome was likely derived from the corresponding cpgenome (Figure 4A,B; Appendix A), occupying 2.2% or 16.3% of the individual mitogenomes. These sequences include five chloroplast genes (*rpoC1*, *ndhB*, *rps7*, *rps19*, and *rpl23*) from both *A. latifolia* and *A. valvata* that have been possibly transferred into their individual mitogenomes (Appendix A).

### 3.5. Phylogenetic Analysis

To study the evolution of the organelle genomes of *A. latifolia* and *A. valvata*, we performed phylogenetic analyses for the organelle genomes including *A. latifolia*, *A. valvata*, and 22 related species. *V. vinifera* and *N. nucifera* were used as the outgroups. In total, we used the nucleotide sequences of 54 common genes (*atpA*, *atpI*, *ndhB*, *petA*, *petL*, *psaB*, *psbB*, *psbE*, *psbI*, *psbL*, *rbcL*, *rpl22*, *rpl33*, *rpoB*, *rps11*, *rps18*, *rps3*, *rps8*, *atpE*, *matK*, *ndhC*, *petB*, *petN*, *psaC*, *psbC*, *psbF*, *psbJ*, *psbM*, *pl14*, *rpl2*, *rpl36*, *rpoC1*, *rps14*, *rps19*, *rps4*, *ycf3*, *atpH*, *ndhA*, *ndhI*, *petG*, *psaA*, *psbA*, *psbD*, *psbH*, *psbK*, *psbT*, *rpl16*, *rpl32*, *rpoA*, *rpoC2*, *rps15*, *rps2*, *rps7*, and *ycf4*) for cpgenome-based phylogenetic analysis (Figure 5). By contrast, we utilized eight common genes (*atp1*, *atp9*, *ccmB*, *ccmC*, *nad6*, *nad9*, *rps4*, and *rps13*) for the mitogenome-based phylogenetic analysis. The trees built with the cpgenomes and the mitogenomes showed that *A. valvata* and *A. arguta*, or *A. latifolia* and *A. eriantha*, were clustered together. The overall structures of the two constructed trees are identical (Figure 5).

### 3.6. Substitution Rates of Protein-Coding Genes

To explore the evolutionary rate of the mitochondrial genes from *A. latifolia* or *A. valvata*, we detected the nonsynonymous substitution rate (dN) and the synonymous substitution rate (dS) for 14 commonly shared protein-coding genes (Figure 6). As a result, there was likely a positive selection on the genes of *ccmB*, *mttB*, and *rps1* due to their dN/dS > 1, while the other genes with lower dN/dS ratios might be under purifying selection (Figure 6). In particular, the *atp1* gene possesses a low dN/dS radio with the lowest variations, implicating that it is a highly conserved gene crucial for the mitogenome functioning.

## 4. Discussion

In the present study, we successfully assembled high-quality mitogenomes of *A. latifolia* and *A. valvata* by combining sequence datasets from Oxford Nanopore long-reads and Illumina short-reads. Sequence assembling unravels that the *A. latifolia* mitogenome possesses a single, circular molecule of 825,163 bp while the *A. valvata* mitogenome consists of two circular molecules in different sizes of 781,709 and 301,558 bp, respectively. The phylogenetic analyses indicated that *A. valvata* and *A. arguta*, or *A. latifolia* and *A. eriantha*, were respectively clustered. Comprehensive analyses focusing on the genomic features, coding genes and repeated sequences, sequence similarity, as well as phylogeny and evolution, were performed.

Sequence assembling revealed *A. valvata* possesses a 31.3% larger size mitogenome with two distinct molecules than that of *A. latifolia* with a single chromosome (Figure 1). However, both mitogenomes contain the same number of 39 protein-coding genes (Appendix A). Interestingly, all the sequenced mitogenomes of six *Actinidia* species and a closely related species *Saurauia tristyla*, regardless of one or two molecules contained, encode 39 proteins, suggesting the number of protein-coding genes in plant mitogenomes is highly conserved [22,42]. In addition, no intron was found in *A. valvata* mitogenome as compared to that of the other six species with 13–17 introns (Appendix A). Previous investigations have revealed that little changes in the number of mitochondrial genes are present between species, but the size of the mitogenome ranges by over a 100-fold, and land plant mitogenomes are characterized with the largest variations. For instance, variations in the mitogenome size are between 13 and 96 kb in algae, while the size of angiosperms’ mitogenomes varies between 200 kilobase and 11 mega base (https://www.ncbi.nlm.nih.gov/genome/browse, accessed on 2 January 2023).

Accumulating evidence implicates that it is not the variations in gene number or intron number contained in these genes that significantly contributes to the size variation or the exceptionally large size of plant mitogenomes. Instead, the majority of the plant mitogenomes contain noncoding sequences with variable sizes [18]. Previous studies have shown that repeated sequences are abundantly distributed in plant mitochondrial genomes, and these sequences are poorly conserved across species and have a high proportion of smaller repeats [43]. In our study, *A. latifolia* mitogenome has 55 SSRs and 23 tandem repeats, strikingly contrasting against the much larger size of 946 SSRs and 120 tandem repeats identified in the *A. valvata* mitogenome, which might contribute to the genome size enlargement (Figure 3A,B; Appendix A). Furthermore, the majority of SSRs (84.56%) from the *A. valvata* mitogenome contain a single-nucleotide repeat unit, indicating that the increased size of the *A. valvata* mitogenome might be partially due to the duplication of short sequences.

Horizontal transfer appears to allow for the acquisition of exogenous fragments, and a proportion of plant mitogenomes could be identified as derivatives from either nuclear, chloroplastic, or viral DNA. However, most of the noncoding sequences are present with unknown origination. Several studies showed a large proportion of the mitogenome is similar to the cpgenome, indicating that DNA transfer events occurred [44,45,46]. Our analysis on the cpgenome and mitogenome sequences suggested that DNA transfer events occurred in the tested cpgenomes. Consistently, our sequence similarity comparison demonstrated that 2.2% or 16.3% sequences in *A. latifolia* or *A. valvata* mitogenome might be derived from their individual cpgenomes (Figure 4A,B; Appendix A). Moreover, construction of phylogenetic relationships using distinct genes sequences from either the cpgenomes or the mitogenomes generated two identical trees with *A. valvata* and *A. arguta*, or *A. latifolia* and *A. eriantha*, clustered in the same individual groups (Figure 5), largely in accordance with previous observations [47]. Consistently, a recent nuclear genome sequence study suggested the divergence of *A. latifolia* and *A. eriantha* was a more recent event than that of other *Actinidia* species [6]. 

We also analyzed the collinearity between the available mitogenomes of closely related species from genus *Actinidia* to inspect DNA rearrangement events within the mitogenome (Figure 2). Dot-plot analysis found that *A. chinensis* has the largest co-linear region corresponding to that of *A. valvata*, while *A. valvata* and *A. latifolia* share the smallest size of co-linear region. The largest co-linear region is about 30 kb. The collinearity blocks for the more distantly related species are relatively smaller probably due to the extensive DNA rearrangements over the past successive generations.

dN/dS analysis is usually used to estimate the selective potentiality of the tested genes. Generally, most mitogenome genes are highly conserved and subjected to neutral evolution under a negative selection. Nevertheless, three genes, i.e., *rps1*, *ccmB*, and *mttB*, are likely under a positive selection due to their dN/dS > 1 (Figure 6). Ribosomal protein S1 (rps1) codes a protein for the small ribosomal subunit S1 that is able to associate with 30S ribosomal RNA [48]. The ccmB gene encodes a member of ccm gene family important for cytochrome c biosynthesis process that has been acquired by plant mitogenome from early prokaryote cells [49]. Several previous studies reported that the ccmB gene has undergone a positive selection in *Scutellaria tsinyunensis* [50] and *Saposhnikovia divaricate* [41]. Howeevr, their biological relevance of these observations remains to be elucidated.

## 5. Conclusions

The size enlargement in *Actinidia* mitogenomes seemed to result from accumulated short sequence duplications and intracellular transfers of the plastid DNA. The number of protein-coding genes remains unchanged, and the majority of the coding genes underwent negative selection, suggesting the mitogenome genes are highly conserved during evolution. Comparative analyses using distinct sequence datasets derived from cpgenomes, mitogenomes, and/or nuclear genomes point to a similar or identical evolutionary relationship among *Actinidia* species. Our study provides important mitochondrial genome resources of *Actinidia* species for understanding organelle genome evolution in plants.

## Figures and Tables

**Figure 1 genes-14-00863-f001:**
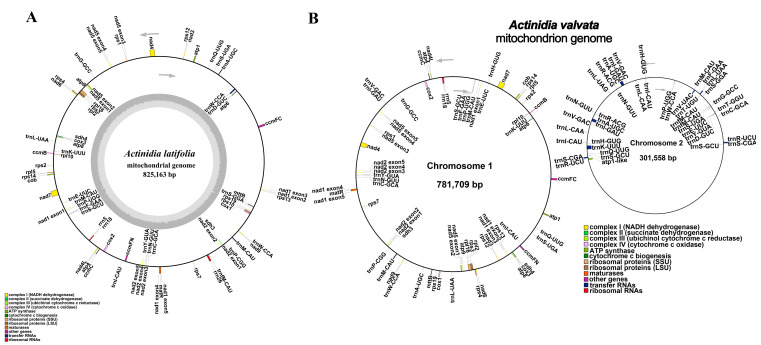
The maps for the circular mitogenomes of *A. latifolia* and *A. valvata*. (**A**) The map for the *A. latifolia* mitogenome. (**B**) The map for the *A. valvata* mitogenome. Genomic characteristics transcribed clockwise or counter-clockwise are indicated on the inside and outside of the circles, respectively. Genes are color-specific depending on their functional classifications. GC content is presented on the inner circle indicated by the dark gray plot.

**Figure 2 genes-14-00863-f002:**
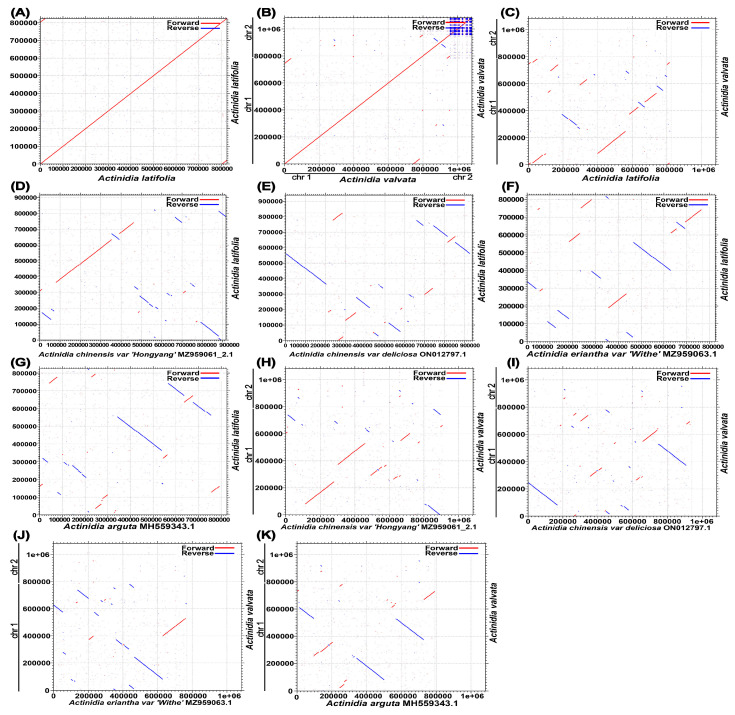
(**A**–**K**) The dotplot diagrams of collinear regions between distinct mitogenomes in closely related species in comparison with *A. valvata* and *A. latifolia*. The red line segment indicates forward direction, and the blue line segment indicates reverse direction.

**Figure 3 genes-14-00863-f003:**
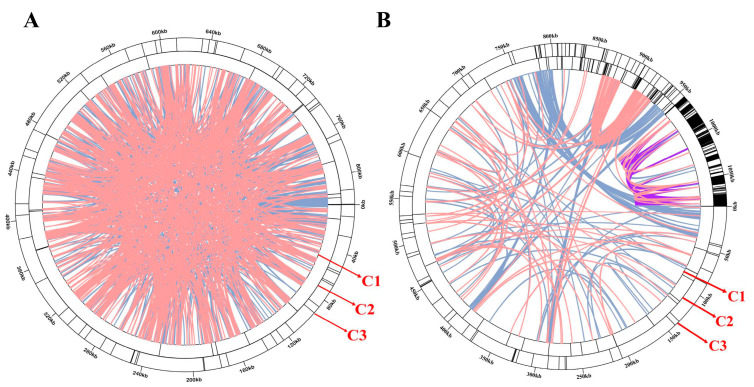
The analyses of repeats from the *A. latifolia* or *A. valvata* mitogenomes. (**A**) The repeated sequences detected in the *A. latifolia* mitogenome. (**B**) The repeated sequences identified from the *A. valvata* mitogenome. C1 circle represents the dispersed repeats connecting with yellow, blue, purple, or pink arcs. C2 circle indicates the tandem repeats as short bars. C3 circle represents the microsatellite sequences detected using MISA (https://webblast.ipk-gatersleben.de/misa/, accessed on 1 January 2023) [26]. The scale on the C3 circle represents 10 kb.

**Figure 4 genes-14-00863-f004:**
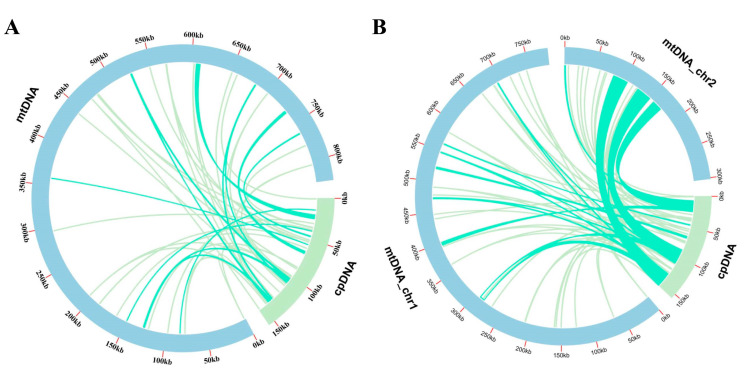
(**A**,**B**) Comparison of the mitogenome sequences derived from individual cpgenomes of the *A. latifolia* and *A. valvata*. The blue and green outer arcs indicate the mitogenome (mtDNA) and cpgenome (cpDNA), respectively, and the inner green arcs represents the homologous DNA fragments. The scale on the outer arcs indicates 20 kb.

**Figure 5 genes-14-00863-f005:**
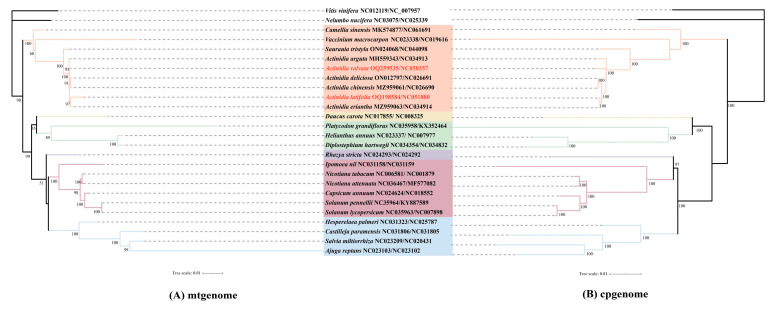
The phylogenetic relationships of the *A. latifolia*, *A. valvata* and other related species. (**A**) Phylogenetic tree of cpgenomes on the basis of nucleotide sequences derived from 54 protein-coding genes. (**B**) Phylogenetic tree depending on the nucleotide sequences of eight protein-coding genes from the available mitogenomes. The sequence from this study is highlighted in red. Phylogenetic tree was constructed using the best evolutionary model “TVM + F + I + G4” and “GTR + F + G4” according to Bayesian Information Criterion (BIC) scores for the cpgenomes and mitogenomes, respectively.

**Figure 6 genes-14-00863-f006:**
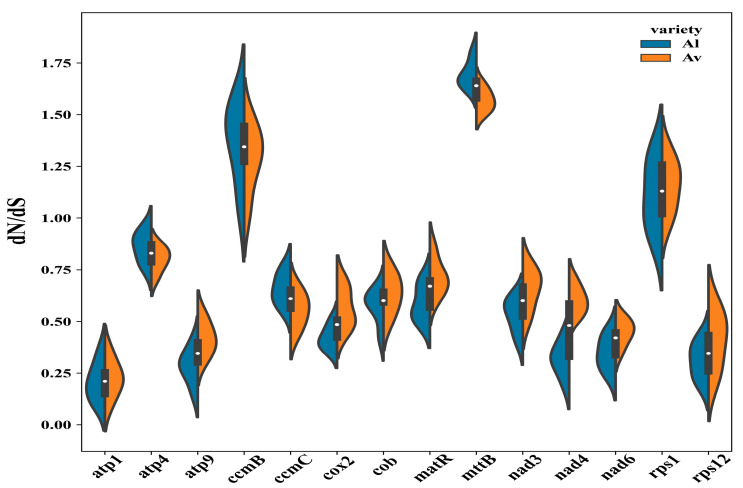
The schematic diagram of dN/dS values for the individual mitochondrial genes in the ten related species. The “X” axis indicates the name of protein-coding genes while the “Y” axis represents the dN/dS values.

## Data Availability

The accession number generated for this study can be found in OQ198584 (*A. latifolia*) and OQ259535-OQ259536 (*A. valvata*). The raw sequencing data have been deposited in NCBI (https://www.ncbi.nlm.nih.gov/ accessed on 26 February 2023) with accession numbers: *A. latifolia*: PRJNA949361, SAMN33942380, SRR23984908 and *A. valvata*: PRJNA949352, SAMN33942102, SRR23984899.

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
