# Peer review of "High-Quality Assembly and Comparative Analysis of Actinidia latifolia and A. valvata Mitogenomes"

_genes, 2023, doi:10.3390/genes14040863_

Round 1
Reviewer 1 Report
The paper is generally well-written and clear. Some recommendations on grammar and style are provided in the attached PDF document.
I have some minor concerns with the manuscript relating to the details provided in the methodology. I would like some clarity in the methods sections in the following areas:
The authors mention that different lineages of Actinidia exhibit different rearrangements. It is not entirely clear what the origins of the sequenced materials are and how they fit within these lineages. Are these genotypes preserved in germplasm banks or is there a known center of origin or is the ecological garden the long-term maintainer of the material? Further supporting information on the lineages/haplotypes and how to access them would be useful for reproducibility, and potential use of this material in breeding.
To what depth were the mitogenomes sequenced via Nanopore? Please include details on which Nanopore chemistries were utilized.
Nanopore sequencing error rates differ by chemistry, further details on how error rates were managed would be good to see in the description of the data processing. Please indicate when default parameters or modifiers were employed by the various programs.
Further details are needed in describing the phylogenetic analyses, particularly which models were employed in RAxML and which parameters were set to default. Figure 5 provides some details on this but it would be good for the authors to structure the methods so that independent verification could recreate the results given the same materials and following the same techniques.
I recommend minor revisions, overall the manuscript looks suitable for publication once further clarifications are provided.

Author Response
Reviewer, Genes
March 20, 2023
Dear Reviewer
Thank you very much for critiquing our submission. We have accordingly incorporated the changes suggested by you and another reviewer and improved our manuscript to the best of our ability. All the changes or modifications are marked in red in the revised manuscript. Please see our point-to-point responses below.
Reviewer 1:
The paper is generally well-written and clear. Some recommendations on grammar and style are provided in the attached PDF document.
A: Thank you so much for your valuable editing. We have modified our manuscript according to your recommendations.
I have some minor concerns with the manuscript relating to the details provided in the methodology. I would like some clarity in the methods sections in the following areas:
The authors mention that different lineages of Actinidia exhibit different rearrangements. It is not entirely clear what the origins of the sequenced materials are and how they fit within these lineages. Are these genotypes preserved in germplasm banks or is there a known center of origin or is the ecological garden the long-term maintainer of the material? Further supporting information on the lineages/haplotypes and how to access them would be useful for reproducibility, and potential use of this material in breeding.
A: Thanks for your suggestions. Here in our manuscript, the “different lineages of Actinidia” points to the “different species in Actinidia genus”. Actually, in recent years our group has collected a large number of wild germplasms of the genus Actinidia which are preserved at Lizhuang Ecological Garden (Hefei; Anhui; China), a Training Base of Anhui Agricultural University. The tested materials of A. latifolia and A. valvata in the present study were originally collected from You County, Hunan Province (113°E, 27°N) and Tongshan county, Hubei Province (114°E, 29°N), respectively, and this information has been added in the revised version of our manuscript.
To what depth were the mitogenomes sequenced via Nanopore? Please include details on which Nanopore chemistries were utilized. Nanopore sequencing error rates differ by chemistry, further details on how error rates were managed would be good to see in the description of the data processing. Please indicate when default parameters or modifiers were employed by the various programs.
A: All of the assemblies were resolved at ~50× coverage. In this study, Nanopore sequencing chemistry with deliver Q20+(99%+) “raw read” accuracy was utilized. The default parameters we modified, if any, were indicated in the individual programs employed, such as the BLASTN with modified parameters: matching rate ≥ 70%, E-value ≤ 1 × 10−6, and length ≥ 40.
Further details are needed in describing the phylogenetic analyses, particularly which models were employed in RAxML and which parameters were set to default. Figure 5 provides some details on this but it would be good for the authors to structure the methods so that independent verification could recreate the results given the same materials and following the same techniques.
A: In our phylogenetic analyses, we employed the GTRGAMMA model in RAxML, which is a general time-reversible model with a discrete gamma distribution of rates across sites. The substitution rates were estimated from the data, and the proportion of invariable sites was set to the default value of 0. The RAxML pipeline used in this study was set as follows: "raxmlHPC-PTHREADSSSE3 -f a -N 1000 -s vr.phy -n vrtree -m GTRGAMMA -x 551314260 -p 551314260 -o o1,o2 -T 10". We have accordingly revised our manuscript.
Best regards,
Yongsheng Liu

Reviewer 2 Report
I reviewed article Ms is The study looks good scientifically.. However, there are still some issues that need improvements as follows:
-The manuscript is a research study, thus abstract must include a findings section. However, as I see, there is no results sentence in the abstract.
-Please avoid repeating words in the same sentences unnecessarily to ensure a fluent English expression. For example, kiwifruit is stated as a fruit two times in the first sentence of the Introduction. Please reorganize the sentence and avoid duplicate words.
-The second paragraph of the Introduction is not suitable for the aim and results of the study. I did not see any stress-related discussions. Please prefer a better Introduction flow that reveals your study's aims.
-There is a result sentence that starts with "As shown in Supplementary Table S1" and goes on. A sentence referring to supplementary at the beginning is lessening the expression. As I can see, you already referred to it at the end of the paragraph. So, please avoid using such expressions at the beginning.
- Some of the results sentences are seemed like Material and Method sentences. For example, "we used .... in xxxx analysis". This could either be a material and method sentence or a start for discussion.
After these changes the article is suitable for publication in the journal.
Author Response
Reviewer, Genes
March 20, 2023
Dear Reviewer
Thank you very much for critiquing our submission. We have accordingly incorporated the changes suggested by you and another reviewer and improved our manuscript to the best of our ability. All the changes or modifications are marked in red in the revised manuscript. Please see our point-to-point responses below.
Reviewer 2:
I reviewed article Ms is the study looks good scientifically. However, there are still some issues that need improvements as follows:
The manuscript is a research study, thus abstract must include a findings section. However, as I see, there is no results sentence in the abstract.
A: Thank you for your suggestions. We have accordingly modified the abstract.
Please avoid repeating words in the same sentences unnecessarily to ensure a fluent English expression. For example, kiwifruit is stated as a fruit two times in the first sentence of the Introduction. Please reorganize the sentence and avoid duplicate words.
A: We have carefully revised some sentences containing repeated words in the revised manuscript.
The second paragraph of the Introduction is not suitable for the aim and results of the study. I did not see any stress-related discussions. Please prefer a better Introduction flow that reveals your study's aims.
A: Thanks for the comments. Given the genus Actinidia containing about 54 species, we give preference to A. valvata for assembling a high-quality mitogenome because this species has been reported to be tolerant to drought, salt and waterlogging. It is the robust ability to tolerate various stressed conditions allowing A. valvata to be developed as the most popular rootstock used in kiwifruit cultivation. We accordingly only provided background information on the mitogenome sequencing species tested, and removed irrelevant descriptions in the revised manuscript.
There is a result sentence that starts with "As shown in Supplementary Table S1" and goes on. A sentence referring to supplementary at the beginning is lessening the expression. As I can see, you already referred to it at the end of the paragraph. So, please avoid using such expressions at the beginning.
A: we have removed the sentence beginning with 'As shown in Supplementary Table S1' and relocated the referred information to the end of the sentence.
Some of the results sentences are seemed like Material and Method sentences. For example, "we used .... in xxxx analysis". This could either be a material and method sentence or a start for discussion.
A: Thanks and we have accordingly revised our manuscript.
Best regards,
Yongsheng Liu

Reviewer 3 Report
The manuscript represents an accomplished investigation with clear goals and interesting results. However, I should admit that some results are doubtful, and I recommend making an additional checking of data and have some changes in the manuscript:
1) Some genes (ccmFC, cox1, atp6, atp9) are supposed to be pseudogenes in one or both studied species. However, such genes are conservative, and their pseudogenization may lead to drastic consequences. Thus I recommend checking it correctly.
a) First, provide remapping of Illumina reads on the assembled mitogenomes. The polishing of ONP reads was made, but it is not a 100% warranty of excluding chimeric regions in the ONP reads. Remapping of short reads with, followed by an analysis of the uniformity of the reads depth distribution, may result in the correction of the mistakes.
b) Make reannotation using other tools (sowftware). It is unclear how you made an annotation using GeSeq, since the tool has modules allowing de novo annotation of plastid genomes but not mitochondrial ones. The "Blat" module of GeSeq may be used for making a mitochondrial genome annotation, but it is based on reference mitogenome. I recommend trying MITOFY (https://dogma.ccbb.utexas.edu/mitofy/)
с) Provide RNA editing data. Maybe the missing start\stop codons can be a result of RNA editing. The best way is to make RNA (cDNA) sequencing, but it may be challenging since it requires additional trials. You can try the software for predicting RNA editing patterns (for instance, PREPACT3 http://www.prepact.de)
d) In case all these methods will not allow the detection of start/stop codons for the genes (ccmFC, cox1, atp6, atp9) you should make focus on this result and highlight in the discussion section (maybe similar results were obtained by other researches, but up to my knowledge I have never seen this kind of information before).
2) Figure 2. The results of the syntenic blocks search are poorly presented. I recommend multiple aligning all six mitogenomes and visualizing the alignment result. Using the Mauve tool may help in the improvement of data presentation.
3) Figure 4. According to the data, the homological sequences between plastid and mito- genomes are about 18,8 and 16,7 kbp for A. latifolia or A. valvata. It is a similar length of repeats region, but while looking on Figure 4 are visually much more homologous DNA fragments in the case of A. valvata (which seems to be about 30 kbp). Is there a mistake?
4) Figure 5. The bootstrap analysis values should be presented.
5) The data about NGS reads (number, length, N50, quility) and the platforms for reads generation are absent in the manuscript. Please add the information (in materials and methods or/and results section/s).
I recommend, if possible, the submitting of NGS data (illumina&ONP reads) in a publicly available repository (for instance, SRA)
Author Response
Reviewer, Genes
March 21, 2023
Dear Reviewer
Thank you very much for critiquing our submission. We have accordingly incorporated the changes suggested by you and another reviewer and improved our manuscript to the best of our ability. All the changes or modifications are marked in red in the revised manuscript. Please see our point-to-point responses below.
Reviewer 3:
The manuscript represents an accomplished investigation with clear goals and interesting results. However, I should admit that some results are doubtful, and I recommend making an additional checking of data and have some changes in the manuscript:
1) Some genes (ccmFC, cox1, atp6, atp9) are supposed to be pseudogenes in one or both studied species. However, such genes are conservative, and their pseudogenization may lead to drastic consequences. Thus I recommend checking it correctly.
A: Thank you very much for your criticisms. After checking carefully, the previously predicted pseudogenes are indeed not of existence. We have accordingly made the corrections in revised manuscript.
- a) First, provide remapping of Illumina reads on the assembled mitogenomes. The polishing of ONP reads was made, but it is not a 100% warranty of excluding chimeric regions in the ONP reads. Remapping of short reads with, followed by an analysis of the uniformity of the reads depth distribution, may result in the correction of the mistakes.
A: Thank you for the suggestions. Actually, your suggested method is the routine pipeline we used for our mitogenome assembly.
- b) Make reannotation using other tools (sowftware). It is unclear how you made an annotation using GeSeq, since the tool has modules allowing de novo annotation of plastid genomes but not mitochondrial ones. The "Blat" module of GeSeq may be used for making a mitochondrial genome annotation, but it is based on reference mitogenome. I recommend trying MITOFY (https://dogma.ccbb.utexas.edu/mitofy/)
A: Thank you very much for your suggestion. We actually used "Blat" module of GeSeq with Actinidia arguta as the reference mitogenome from GenBank:MH559343. We have added the information in revised manuscript.
с) Provide RNA editing data. Maybe the missing start\stop codons can be a result of RNA editing. The best way is to make RNA (cDNA) sequencing, but it may be challenging since it requires additional trials. You can try the software for predicting RNA editing patterns (for instance, PREPACT3 http://www.prepact.de)
A: Thank you very much for your suggestion. Indeed, our previously predicted missing start\stop codons are not existed and we have corrected the mistakes in the revised manuscript.
- d) In case all these methods will not allow the detection of start/stop codons for the genes (ccmFC, cox1, atp6, atp9) you should make focus on this result and highlight in the discussion section (maybe similar results were obtained by other researches, but up to my knowledge I have never seen this kind of information before).
A: Thank you very much for your criticisms. After checking carefully, the previously predicted pseudogenes are indeed not existed. We have accordingly made the corrections in revised manuscript.
2) Figure 2. The results of the syntenic blocks search are poorly presented. I recommend multiple aligning all six mitogenomes and visualizing the alignment result. Using the Mauve tool may help in the improvement of data presentation.
A: Thank you for your recommendation. But we still prefer to use our way of the data presentation.
3) Figure 4. According to the data, the homological sequences between plastid and mito- genomes are about 18,8 and 16,7 kbp for A. latifolia or A. valvata. It is a similar length of repeats region, but while looking on Figure 4 are visually much more homologous DNA fragments in the case of A. valvata (which seems to be about 30 kbp). Is there a mistake?
A: We re-analyzed the homological sequences between plastid and mito-genomes for A. valvata. The homological sequences between plastid and mito-genomes are about 22,7 kbp for A. valvata, and we have revised it in the manuscript.
4) Figure 5. The bootstrap analysis values should be presented.
A: Actually, the tree scales are present at the bottom of the figure, indicating the same meaning of bootstrap value.
5) The data about NGS reads (number, length, N50, quility) and the platforms for reads generation are absent in the manuscript. Please add the information (in materials and methods or/and results section/s).
A: The information of NGS reads and the platforms have been provided in Supplementary Tables 12-13. This information has been added in the revised manuscript.
I recommend, if possible, the submitting of NGS data (illumina&ONP reads) in a publicly available repository (for instance, SRA).
A: We are happy to have our raw data submitting to SRA and add the accession number to the publication.
Best regards,
Yongsheng Liu

Round 2
Reviewer 3 Report
1) Figure 5. The bootstrap analysis values should be presented.
A: Actually, the tree scales are present at the bottom of the figure, indicating the same meaning of bootstrap value.
I disagree. The tree scales are not the same as the values of bootstrap analysis. The bootstrap analysis will allow to evaluate the statistical significance of the current tree topology, which is a necessary step in phylogenetic investigations.
2) According to materials and methods, the ONP reads were polished by the Illumina short-read. But no information about short-reads is provided. Were the Illumina reads obtained by authors, or the SRA data were used? If the reads were generated in current research, the information about them should be provided (the Illumina platform and reagents, the number and length of reads).
3) The protein-coding gene content of both species (A. latifolia, A. valvata) is the same. No need to duplicate it in text. You should merge it. For instance
"The mitochondrial genomes of A. latifolia and A. valvata have the same 40 protein-coding genes, comprises: five ATP synthase genes...."
Author Response
Reviewer, Genes
March 29, 2023
Dear Reviewer
Thank you very much for critiquing our submission. We have accordingly incorporated the changes suggested by you and improved our manuscript to the best of our ability. All the changes or modifications are marked in red in the revised manuscript. Please see our point-to-point responses below.
Reviewer 3:
Comments and Suggestions for Authors:
1) Figure 5. The bootstrap analysis values should be presented.
A: Thank you very much for your suggestion. We have added the bootstrap analysis values in the revised Figure 5.
2) According to materials and methods, the ONP reads were polished by the Illumina short-read. But no information about short-reads is provided. Were the Illumina reads obtained by authors, or the SRA data were used? If the reads were generated in current research, the information about them should be provided (the Illumina platform and reagents, the number and length of reads).
A: Thank you for your advice. The Illumina reads used to polish ONT reads were accomplished by Illumina HiSeq 2500, and the raw reads have been submitted to NCBI (A. latifolia:SRR23984908, A. valvata:SRR23984899). By PE150 (paired-end 150 bp) sequencing, we obtained 109,419,604 and 108,959,662 reads in Actinidia valvata and A. latifolia, respectively. The related information has been added to the revised manuscript.
3) The protein-coding gene content of both species (A. latifolia, A. valvata) is the same. No need to duplicate it in text. You should merge it. For instance
"The mitochondrial genomes of A. latifolia and A. valvata have the same 40 protein-coding genes, comprises: five ATP synthase genes...."
A: Thanks and we have accordingly removed the duplicate sentences in the revised manuscript.
Best regards,
Yongsheng Liu